# Corrosion Behavior and Mechanical Performance of 7085 Aluminum Alloy in a Humid and Hot Marine Atmosphere

**DOI:** 10.3390/ma15217503

**Published:** 2022-10-26

**Authors:** Jianquan Tao, Lin Xiang, Yanping Zhang, Zude Zhao, Yan Su, Qiang Chen, Jipeng Sun, Bo Huang, Feifei Peng

**Affiliations:** Southwest Technology and Engineering Research Institute, Chongqing 400039, China

**Keywords:** 7085 aluminum alloy, humid and hot marine atmosphere, outdoor exposure, corrosion behavior, mechanical performance

## Abstract

This work analyzed changes in the corrosion morphology and mechanical performance of 7085 aluminum alloy after outdoor exposures for different times in a humid and hot marine atmospheric environment. After one month of exposure, a pronounced corrosion of the alloy was observed. The corrosion product was mainly Al(OH)_3_, and the corrosion features were mainly pitting corrosion and intergranular corrosion (IGC). With the exposure time extended from 6 months to 12 months, the IGC depth increased from 114 μm to 190 μm. After a 1-year outdoor exposure in a humid and hot marine environment, the alloy’s ultimate strength and yield stress remained nearly unchanged, while its elongation and area reduction decreased from the original 6% and 9%, respectively, to 3% and 5%. Moreover, the reasons for IGC and its effect on the mechanical performance was analyzed.

## 1. Introduction

7xxx series aluminum alloys have been widely applied in the aviation and aerospace industry due to their excellent properties, such as a low density, high specific strength, good processing performance, and so on [1,2,3]. Among 7xxx series aluminum alloy, 7085 alloy possesses an outstanding and advantageous status because of an extra-good corrosion resistance and low quenching sensitivity [4,5,6]. Thus, many studies have focused on its microstructure and mechanical properties [7,8,9,10]. For aluminum alloy components, they frequently operate in coastal regions, with a high salt spray presence, high humidity, and hot atmosphere. The corrosive effects of such an environment include IGC, pitting corrosion, and stress corrosion of special equipment, significantly deteriorating their service life and safety [11,12,13,14]. Therefore, clarifying the corrosion behavior and mechanical strength variation of the alloy under study in a corrosive environment, especially in an actual marine atmosphere, is quite topical.

Recently, some work on the corrosion behavior of aluminum alloys has been performed [15,16,17,18,19,20]. Zhao et al. studied the atmospheric corrosion behavior and mechanism of 7A85 aluminum alloy in Qingdao industrial-marine atmosphere and concluded that pitting corrosion and IGC caused significant deterioration of the mechanical properties of 7A85 aluminum alloy [17]. Peng et al. studied the corrosion behavior of extruded 6061 aluminum alloy exposed to a simulated Nansha marine atmosphere for 40 days and revealed severe IGC and intragranular corrosion on the alloy sample surface [19]. Song et al. studied the fatigue damage evolution in 7075 alloy and found that the precorrosion in a standard exfoliation corrosion solution leads to a significant decrease of the fatigue property. According to previous studies, it is obvious that aluminum alloys experience serious corrosion in a corrosive environment, resulting in the deterioration of mechanical properties.

Therefore, this work aims to study the corrosion behavior and mechanical performance of 7085 alloy in the South China Sea. Correspondingly, outdoor exposure tests in Wanning, Hainan Province of China, were designed. After outdoor exposure, the macroscopic corrosion features, microstructure, and tensile properties were analyzed. The mechanisms of the 7085 alloy’s corrosion in actual humid and hot marine atmospheres and its effects on tensile properties were also discussed.

## 2. Experimental Section

The material under this work was 7085 aluminum alloy in the form of a forging billet. Its chemical composition is listed in Table 1. The alloy was solution-treated at 470 °C for 6 h, followed by a 5% cold compression. Finally, it was artificially two-step-aged at 120 °C for 6 h and 160 °C for 10 h and named the T7452 treatment. The tabular samples with a size of 100 mm × 50 mm × 3 mm and dumbbell-shaped tensile samples with a diameter of 10 mm were cut from the forging billet, which was placed in a humid and hot marine atmospheric environment (in Wanning city, Hainan Island, China) for outdoor exposure tests. The average air temperature, relative humidity, and Cl^−^ deposition rate were about 23.9 °C, 87.6%, and 14.5875 mg/(m^2^·d), respectively [21]. The samples were firmly installed on the exposure test frame and then inclined at 45° to the horizontal surface, as shown in Figure 1. To observe the cross-section morphology and mechanical performance, the samples were analyzed after exposure for 1, 6, and 12 months.

The microstructure was analyzed by optical microscopy (OM), X-ray diffraction (XRD), scanning electron microscopy (SEM), and transmission electron microscopy (TEM). The sample etchant for the metallographic analysis was Keller’s reagent, consisting of 95 mL DI water, 2.5 mL HNO_3_, 1.5 mL HCl (36%), and 1.0 mL HF. The metallographic microstructure was observed by OM (Leica DFC320) and SEM (JEOL JSM-6390A), and the energy dispersive spectrometer (EDS) of the SEM was used for the compositional analysis. The SEM was executed in secondary electron mode, and its voltage was 20 KV. The varieties of the second phases were analyzed by XRD (D/MAX 2200 PC). The X-ray source was a Cu (K_α_) target with a wavelength of λ = 1.5406 Å under the conditions of 40 kV tube pressure, 30 mA tube flow, 0.02° scan step size, and 10°~90° scanning range. The microscopic structural analysis was performed by TEM (Tecnai F30 G2) with an accelerating voltage of 300 kV. The preparation method is as follows: a 1 mm square piece was cut from the surface of the sample by the wire cutting method. Then, the specimen was polished using 600#~1500# sandpaper to reduce the thickness to less than 0.05 mm, and double-spray electropolishing technology was used to thin the perforation. The double-spray electrolyte was HClO_4_:C_4_H_9_OH:CH_3_OH = 6:34:60 (volume ratio) cooled by liquid nitrogen, and the electrolysis temperature was controlled at approximately −30 °C. The tensile performance at room temperature was tested by a universal testing machine, and the tensile speed was 2 mm/min. Each measured result takes the average of 3 samples.

## 3. Results and Discussion

### 3.1. Original Microstructure

Figure 2 shows the original microstructure of the 7085 alloy. As observed, the microstructure is mainly composed of α(Al) grains and granular secondary phases. The α(Al) grains are equiaxed, and the grain size is small and uniform. According to the SEM image, the second phase particles precipitated continuously and distributed densely at the grain boundaries of the matrix. Moreover, the size of second phase particles is not uniform. The XRD analysis results of the original sample are shown in Figure 3, and the results indicate that the second phase particles belong to the MgZn_2_ phase.

### 3.2. Macroscopic Corrosion Features

Figure 4 shows the macroscopic morphology of the 7085 alloy exposed to the humid and hot marine atmospheric environment with different exposure times. It is obvious that corrosion behaviors were observed on the samples surface after exposure to a humid and hot marine atmospheric environment for one month. Specifically, a large number of clustered gray products (approximately 30% in area) were observed on the sample surface, as shown in Figure 4a. With a prolonged exposure time from 1 month to 6 months and to 12 months, as shown in Figure 4a,b, the area fraction of the corrosion product on the samples’ surface did not increase significantly.

Figure 5 shows the morphology and XRD map of the corrosion product of the 7085 alloy in the humid and hot marine environment. According to Figure 5a, the corrosion product on the specimens’ surface was relatively dense, which could protect the matrix well during outdoor exposure. This is the reason why the area fraction of the corrosion product did not obviously increase with a prolonged exposure time. The XRD result indicated that in humid and hot marine environments, the corrosion surface of the present alloy was mainly composed of an Al matrix, Al(OH)_3_, and AlCl_3_. Thus, one can speculate that the corrosion products were mainly Al(OH)_3_ and AlCl_3_. As shown in Figure 5a, the EDS result indicated that the main elements of the corrosion product were O and Al, and the atomic percentage was approximately 3:1. It could also be demonstrated that the corrosion product of the 7085 alloy was mainly Al(OH)_3_. The marine environment, with high salt fog and a high Cl^-^ concentration, will accelerate the corrosion, resulting in the occurrence of AlCl_3_.

### 3.3. Microscopic Corrosion Features

To reveal the corrosion mechanism of the 7085 alloy, its microscopic corrosion features were analyzed. Figure 6 shows the cross-sectional morphology of the 7085 alloy after outdoor exposure. In a humid and hot marine environment, the 7085 alloy exhibited significant pitting corrosion and IGC. Meanwhile, pitting corrosion was mainly on the surface of the section, and IGC was mainly in the interior of the alloy. However, IGC and pitting corrosion were connected to each other. Moreover, IGC spread along the grain boundaries of fine equiaxed Al grains, and the corrosion was mainly intergranular corrosion. With the exposure time extended from 6 months to 12 months, as shown in Figure 6c, the IGC depth increased from 114 μm to 190 μm. Accordingly, the average weight loss increased from 0.0231 g to 0.0294 g. Thus, after outdoor exposure for 12 months, the corrosion rate of 7085 alloy was calculated, and the value was 2.7013 g/(m^2^ a).

Figure 7 depicts the TEM results of the second phase of the 7085 alloy. As shown in Figure 7a, the second phase was densely distributed at the grain boundary, which was consistent with the SEM observation. Meanwhile, a large number of dispersed second phases were also found in the Al matrix, and their size was rather small at the nanoscale. The electron diffraction pattern and high-resolution analysis in Figure 7b–d revealed that the second phase at the grain boundary was the η (MgZn_2_) phase with a close-packed hexagonal structure, while the fine second phase in the grain was a metastable (MgZn_2_) phase with a simple hexagonal structure. Herein, the lattice parameters of the η phase were *a* = *b* = 0.522 nm and *c* = 0.857 nm, showing an incoherent relationship with the Al matrix, with a directional relationship of [−1–121]_η_∥[1–10]_α-Al_. The lattice parameters of the $\eta^{\prime }$ phase were *a* = *b* = 0.505 nm and *c* = 1.402 nm, showing a semicoherent relationship with the Al matrix.

The corrosion behavior of alloys is known to be closely related to their microstructure. When grain boundary precipitates of the alloy distribute continuously, the alloy has a higher sensitivity to IGC [22,23,24]. In 7xxx series aluminum alloys, the potentials of the η (MgZn_2_) phase and Al matrix were −0.86 and −0.68 V, respectively [25]. Hence, relative to the Al matrix, the η phase easily became the anodic dissolved phase, which was preferentially corroded in the humid and hot marine environment. Therefore, the continuous corrosion of the grain boundary η phase provided the conditions for IGC. A further analysis of the microscopic characteristics of the 7085 alloy is shown in Figure 8. A high, geometrically necessary dislocation density exists in the grain boundary and the area of the intergranular crack, indicating that there was a stress concentration in the grain boundary, which would promote IGC. In summary, there are two reasons for the IGC of the 7085 alloy. One is that there is a continuous anodic dissolution of the η phase at the grain boundary. During the corrosion process, the η phase is continuously corroded and consumed, and the intergranular crack expands. The other reason is the high stress concentration in the α-Al crystal boundary, which promotes the progress of IGC.

### 3.4. Variation in Mechanical Performance

Figure 9 shows the tensile performance of the 7085 alloy after outdoor exposure for different times in humid and hot marine environments. The original ultimate strength and yield stress were 510 MPa and 462 MPa, respectively. After exposure for 12 months, it is obvious that the ultimate strength and yield stress of the 7085 alloy did not change significantly. However, after one year of exposure, the alloy’s elongation and area reduction began to decrease (from the original 6% and 9%, respectively, to 3% and 5%). This pattern differed from that of the 7085 alloy fatigue strength, which decreased more than 90% after 3 months of outdoor exposure in a marine environment [26]. Additionally, the ultimate strength variation of the present alloy was slower than that of 2xxx aluminum alloys, which decreased by more than 10% after outdoor exposure in a marine environment [27]. Combined with the analysis of the corrosion features, the main corrosion mechanism of the 7085 alloy was IGC, which formed a large number of intergranular cracks inside the microstructure. With a prolonged corrosion time, the IGC depth increased. The cracks were also observed on the corrosion surface and fracture surface, as shown in Figure 10. However, the tensile performance, especially the strength, did not change significantly, indicating that the tensile performance of the 7085 alloy was not sensitive to the IGC behavior and that only when the IGC reached a certain depth would it lead to a tensile performance reduction.

## 4. Conclusions

The results obtained made it possible to draw the following conclusions:After exposure to a humid and hot marine environment for one month, Al alloy 7085 exhibited an obvious corrosion behavior, and the corrosion product was relatively dense, mainly Al(OH)_3_. With a prolonged exposure time, the corrosion product did not increase significantly.In a humid and hot marine atmospheric environment, the main corrosion feature of the 7085 alloy was IGC, resulting from the continuous anodic dissolution of the η (MgZn_2_) phase and the stress concentration at the α-Al grain boundary. With the exposure time extended from 6 months to 12 months, the IGC depth increased from 114 μm to 190 μm. Accordingly, the average weight loss increased from 0.0231 g to 0.0294 g.After a 1-year outdoor exposure of the 7085 alloy in a humid and hot marine atmospheric environment, its ultimate tensile strength and yield stress values did not change significantly. However, the elongation and area reduction decreased from the original 6% and 9%, respectively, to 3% and 5%. The variation of the tensile performance is mainly ascribed to intergranular cracks.

## Figures and Tables

**Figure 1 materials-15-07503-f001:**
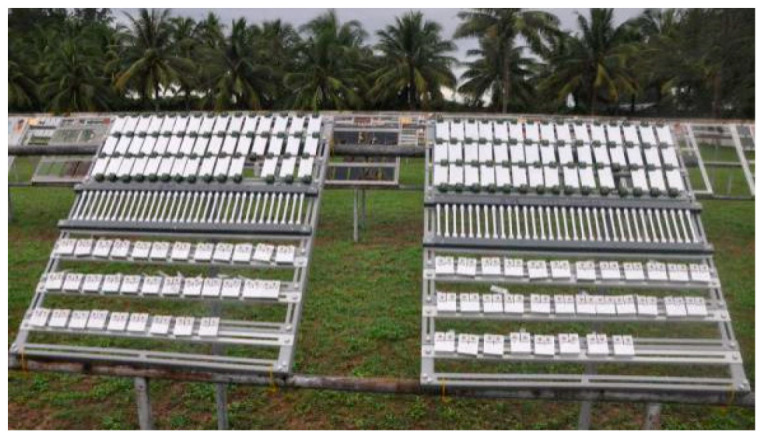
Outdoor exposure test in a humid and hot marine atmospheric environment.

**Figure 2 materials-15-07503-f002:**
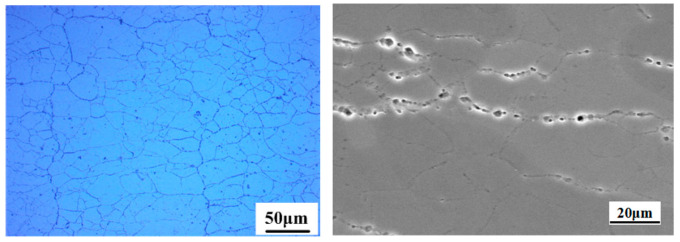
Original microstructure of the present alloy.

**Figure 3 materials-15-07503-f003:**
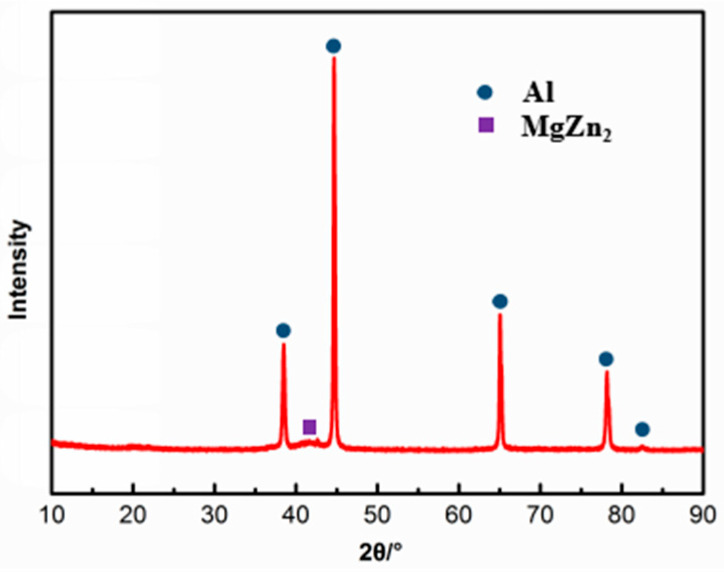
Original XRD spectra of the present alloy.

**Figure 4 materials-15-07503-f004:**
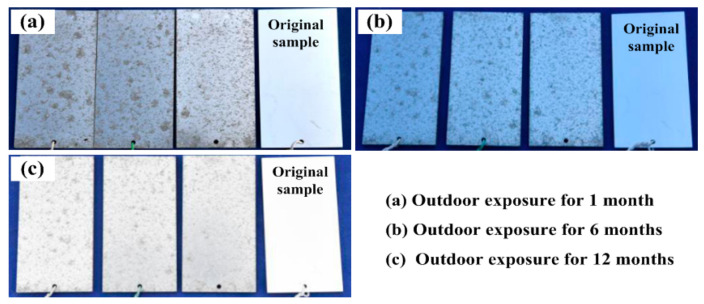
Macroscopic morphology of the 7085 alloy exposed to a humid and hot marine atmospheric environment: (**a**) outdoor exposure for 1 month; (**b**) outdoor exposure for 6 months and (**c**) outdoor exposure for 12 months.

**Figure 5 materials-15-07503-f005:**
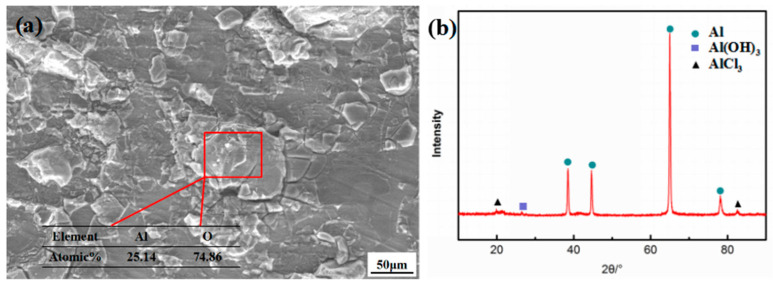
7085 alloy corrosion product morphology and XRD spectra in the humid and hot marine environment: (**a**) corrosion product morphology and (**b**) XRD map.

**Figure 6 materials-15-07503-f006:**
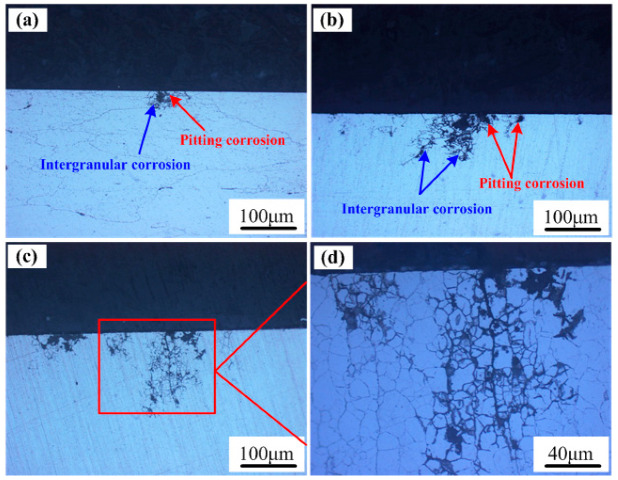
The cross-sectional morphology of the 7085 alloy exposed in the humid and hot marine environment for (**a**) 1 month, (**b**) 6 months, and (**c**,**d**) 12 months. The red rectangle in (**c**) is magnified in (**d**).

**Figure 7 materials-15-07503-f007:**
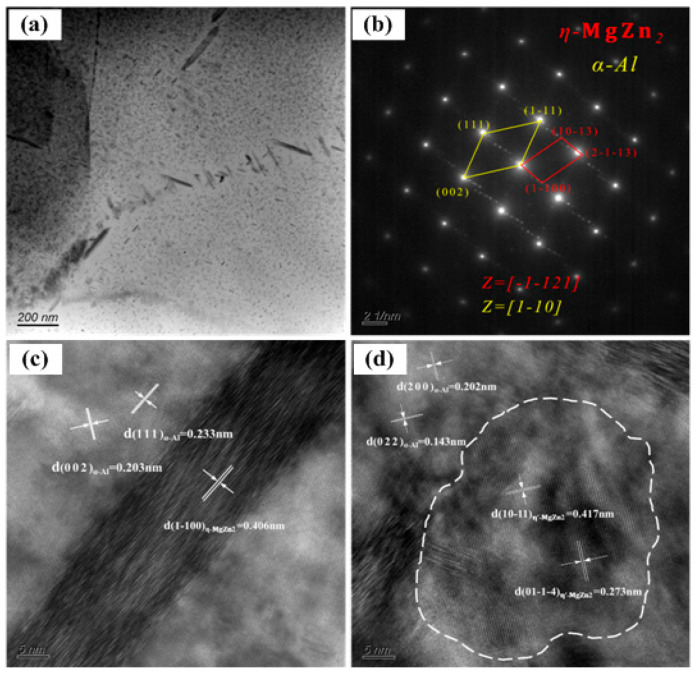
TEM images of the second phase: (**a**) bright field image, (**b**) electron diffraction pattern of the second phase at the grain boundary, and high-resolution images of the (**c**) grain boundary and (**d**) grain interior.

**Figure 8 materials-15-07503-f008:**
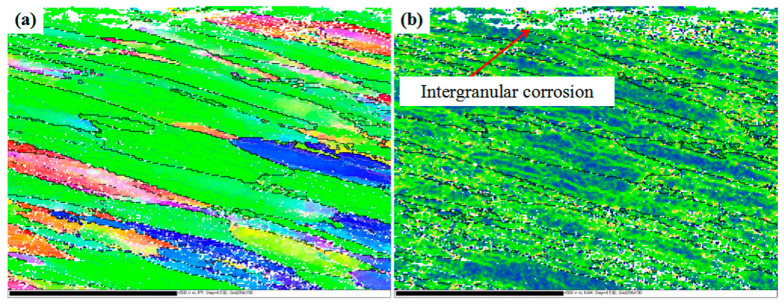
The EBSD map of the 7085 alloy exposed in a humid and hot marine environment: (**a**) IPF; (**b**) KAM.

**Figure 9 materials-15-07503-f009:**
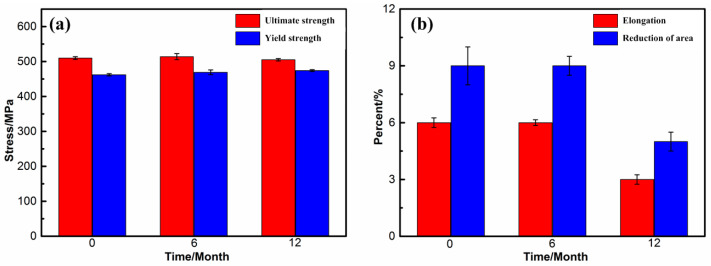
Tensile performance of the 7085 alloy exposed for different times in humid and hot marine environments: (**a**) ultimate strength and yield stress; (**b**) elongation and area reduction.

**Figure 10 materials-15-07503-f010:**
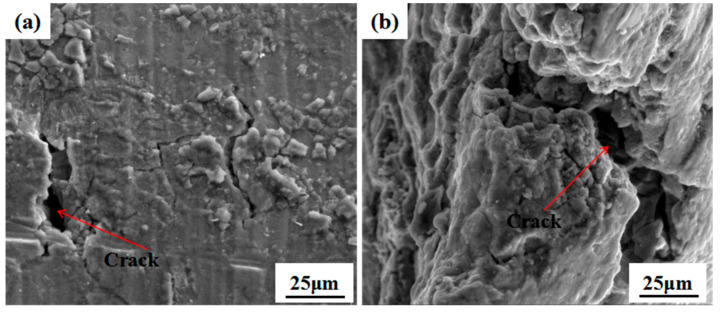
Cracks of the 7085 alloy after outdoor exposure in humid and hot marine environments: (**a**) corrosion surface; (**b**) fracture surfaces.

**Table 1 materials-15-07503-t001:** Chemical composition (wt.%) of 7085 aluminum alloy.

Element	Cu	Zn	Mg	Zr	Si	Ti	Fe	Cr	Mn	Al
Weight fraction	1.77	7.88	1.54	0.11	0.01	0.027	0.026	<0.01	<0.01	Bal.

## Data Availability

Not applicable.

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
