# Peer review of "Corrosion Behavior and Mechanical Performance of 7085 Aluminum Alloy in a Humid and Hot Marine Atmosphere"

_materials, 2022, doi:10.3390/ma15217503_

Round 1

Reviewer 1 Report

In the present work, the 7085 Aluminum alloy has been outdoor tested for its susceptivity to corrosion in a humid and hot-marine environment.   The alloy was initially heat treated according to the T7452 treatment. The mechanical properties, ultimate tensile strength and the yield strength, the elongation, and the cross-area reduction of the sample were evaluated before and after the sample exposure to the marine environment under predetermined duration of the outdoor exposure, i.e., after 1, 6, and 12 months. Furthermore, microstructures of the samples were examined after the different exposure of the samples to the marine environment. The microstructure was analyzed by optical microscopy (OM), X-ray diffraction (XRD), scanning electron microscopy (SEM), and transmission electron microscopy (TEM).

Examinations of the OM show that the original microstructure of the alloy consists of α(Al) grains and second phase particles at the grain boundaries. The XRD analysis indicates that the second phase particles are the η (MgZn2) phase.

Also, examination of morphology, i.e., surface, of the samples after 1,6,12 months of the exposure demonstrated that the corrosion product of the 7085 alloy was mainly Al(OH)3.   On the contrary, Examinations of the cross-section area of the samples after 1,6, and 12 months of the exposure indicated that mainly intergranular corrosion (IGC) had occurred in a progressive manner as a function of the elapsed time of exposure.  In the meantime, the TEM examinations & electron diffraction have confirmed the SEM examinations and XRD analysis that the second phase particles in the grain boundaries are mainly the η MgZn2 phase.

The progressive IGC took place due to possible of two reasons.

1-the potential difference between the matrix phase (-0.68V) and the second particle phase (-0.86V) which made the grain boundaries are anodic areas compared to the matrix in a humid environment, due to a galvanic coupling mechanism. 

2-The present of a high dislocation density in the matrix, as indicated by TEM examinations, compared to the grain boundaries made the grain boundaries regions of a high stress-concentrated for initiation of microcracks leading to IGC. 

Also, the mechanical properties on one hand didn't significantly change as far as the ultimate tensile strength and the yield strength of the samples after the exposure to the outdoor marine environment. On the other hand, the elongation and the cross-area reduction of the sample were found to reduce from 6-9% and 3 to 5 %, respectively, after the exposure to the marine environment.   

Reviewer 2 Report

Reviewers' comments:

Manuscript Number: materials-1981718

Title: Corrosion behavior and mechanical performance of 7085 aluminum alloy in a humid and hot marine atmosphere.

Comments: 

This work concerns the Corrosion behavior and mechanical performance of 7085 aluminum alloy in a humid and hot marine atmosphere. Important points are missing and there are some points that should be revised or corrected. Some important points are mentioned hereafter.

- Abstract looks very general and not informative. In abstract authors should mention should mention the values of results and importance of research work in one or two sentences.

- Introduction part is lagging the previous studies on. Literature review is not updated in the manuscript. More recent literature need to be added.

- There is no information about the novelty of the prepared materials. There is no information about it in introduction section.

- Give more detail for the scanning electron microscopy (SEM) analysis.

- In part SEM: how the energy of the accelerator beam used?

- Figure 2. Original microstructure of the present alloy – not clear make clear.

- Figure 5. - Not clear make clear.

- 3.1. Macroscopic corrosion features – section should be improved.

- Figure 8. The EBSD map of the 7085 alloy exposed in a humid and hot marine environment: (a) IPF; (b) KAM - Not clear make clear.

- The results and discussion section can be re-structured to make it easier for the reader.

- 3.3. Variation in mechanical performance - section should be improved.

- Main findings should also be provided in conclusions.

- Author should add more recent references.

- Make all references in same format for volume number, page numbers and journal name, because it is difficult to searching and reading.

- The English of the manuscript needs to be improved.

Based on these, I advise the authors to rectify the above mentioned errors and we hope to re-evaluate the revised manuscript.

Author Response

Please see the attcachment.

Reviewer 3 Report

In the paper, the authors studied the atmospheric corrosion and mechanical properties of a 7085 aluminum alloy. The paper is interesting and novel. However, it is relatively short (8 pages) as it lacks some experimental details and a deeper discussion. It is publishable subject to revision.

1.The authors speak of high temperatures (line 6 of the Introduction); however, the alloy was exposed to ambient temperature. Please, specify the exposure conditions of your experiments (average temperature, humidity, etc.).

2.Figures 2 and 3 are results. They should be moved to the Results and Discussion section.

3.The corrosion behavior has been studied by observing the microstructure (Fig. 6). It would help to quantify the results, i.e., to show the weight loss/weight gain of the alloy and calculate the corrosion rate.

4.Experimental stress-strain curves are missing in the manuscript.

5.The authors should show the fracture surfaces after mechanical testing to confirm the suggested intergranular cracking.

6.The discussion is very short. The results (corrosion rate, UTS) should be compared with those of previously studied aluminum alloys.

Round 2

Reviewer 2 Report

The authors revised the manuscript according to the reviewers' comments.

Reviewer 3 Report

Authors answered my comments. The current version is acceptable.